# Integration of Membrane Processes for By-Product Valorization to Improve the Eco-Efficiency of Small/Medium Size Cheese Dairy Plants

**DOI:** 10.3390/foods10081740

**Published:** 2021-07-28

**Authors:** Antónia Macedo, José Bilau, Eunice Cambóias, Elizabeth Duarte

**Affiliations:** 1Polytechnic Institute of Beja, Rua Pedro Soares, Campus do IPBeja, 7800-295 Beja, Portugal; jose.bilau@ipbeja.pt (J.B.); eunice_camboias@hotmail.com (E.C.); 2LEAF—Linking Landscape, Environment, Agriculture and Food, Instituto Superior de Agronomia, University of Lisbon, Tapada da Ajuda, 1349-017 Lisboa, Portugal; eduarte@isa.utl.pt; 3Center for Advanced Studies in Management and Economics of the University of Évora (CEFAGE-UE), 7005-869 Évora, Portugal; 4Instituto Superior de Agronomia, University of Lisbon, Tapada da Ajuda, 1349-017 Lisboa, Portugal

**Keywords:** second sheep cheese whey, goat cheese whey, ultrafiltration/diafiltration, process design, cost–benefit analysis

## Abstract

Goat and second cheese whey from sheep’s milk are by-products of the manufacture of goat cheeses and whey cheeses from sheep. Due to their composition which, apart from water—about 92%—includes lactose, proteins, fat, and minerals, and the elevated volumes generated, these by-products constitute one of the main problems facing to cheese producers. Aiming to add value to those by-products, this study evaluates the efficiency of ultrafiltration/diafiltration (UF/DF) for the recovery of protein fraction, the most valuable component. For a daily production of 3500 and using the experimental results obtained in the UF/DF tests, a membrane installation was designed for valorization of protein fraction, which currently have no commercial value. A Cost–Benefit Analysis (CBA) and Sensitivity Analysis (SA) were performed to evaluate the profitability of installing that membrane unit to produce three new innovative products from the liquid whey protein concentrates (LWPC), namely food gels, protein concentrates in powder and whey cheeses with probiotics. It was possible to obtain LWPC of around 80% and 64% of crude protein, from second sheep cheese whey and goat cheese whey, respectively. From a survey of commercial values for the intended applications, the results of CBA and SA show that this system is economically viable in small/medium sized cheese dairies.

## 1. Introduction

Cheese whey nutritional composition has increasingly aroused the interest for its reuse, because in addition to reducing its environmental impact, it can also contribute to various benefits in the food industry, such as: improved texture; enhancement of flavor and color; increased stability; emulsifying function; gelling ability and improvement of the nutritional value of food, mainly attributed to the excellent quality of whey proteins [1,2,3]. Since in cheese whey about 92% is water [4], separation and concentration of the nutritional components is fundamental for its valorization. Fractionation of cheese whey have been used by the dairy industry to obtain different commercial products most of them based on bovine whey proteins, particularly: dry protein concentrates with protein contents between 35% and 80% from ultrafiltration and protein isolates with protein concentrations greater than 80%, obtained by ultrafiltration/diafiltration or ion chromatography [5,6,7,8]. However, when dealing with membrane processes, the major problems that can happen are related to fouling of membranes or to concentration polarization phenomena, which are responsible not only for the decline of permeation fluxes, but also for changing the separation properties of membranes [9,10]. The extent of this phenomena depends on several factors, namely composition of the cheese whey, membrane characteristics, pre-treatments and monitoring of operating parameters, such as transmembrane pressure, feed flow circulation rate, and temperature [9]. The major foulants of UF membranes are whey proteins (mainly β-lactoglobulin and α-lactalbumin), minerals, especially calcium and phosphate and residues from upstream processing, such as, curd, residuals lipids, caseinomacropeptide, enzymes, and microorganisms [11]. With respect to membrane fouling by whey proteins, it has been proved that a more hydrophilic, softer, and neutrally charged membrane results in a low fouling potential [12,13]. Mineral precipitation by calcium salts is more likely at high pH, temperature, and calcium concentration, although these phenomena can be more important in UF permeates, due to the buffer capacity of whey proteins [14]. Despite the globular fat is not usually identified as UF membrane foulant in the case of cheese whey, a high lipid content and/or free fat can however cause severe fouling. Therefore, the skimming of cheese whey should be performed, to avoid membrane fouling. The use of the pre-concentration of the desired macrosolute, followed by dilution mode (DF) in ultrafiltration to purify whey protein concentrates enhance the degree of separation between membrane-retained species and membrane-permeable species [15]. Whey proteins are sources of peptides biologically active with important functions in human health—including antihypertensive, antioxidant, and antimicrobial activities—as opioids and with ability to decrease cholesterol levels in the body [16,17,18]. However, this bioactivity is latent until released and activated during gastrointestinal digestion or food processing, due to fermentation processes and/or acidification [17,19]. Therefore, the use of whey protein concentrates from ultrafiltration to produce new fermented products such as acid dairy gels yogurt or dessert type, by bacterial fermentation or chemical acidification not only allows for the extension of shelf life of these products, but also can give significant health benefits, such as increase the amount of easily digestible amino acids, control of intestinal flora, helps for lactose digestion in intolerant individuals [20,21,22,23,24]. The production of artisanal sheep, goat cheeses and whey cheeses, these last obtained from thermal precipitation of whey proteins, is generally carried out in small/medium size cheese dairy plants. The goat cheese whey and second cheese whey from sheep milk is provided free of charge to animal producers, mainly for feeding pigs, or discharged into septic tanks that must be cleaned periodically or still conducted, for water treatment plants. The integration of membrane processes for cheese whey recovery and further application in the development of new products that can diversify the cheesemaking activity, eventually increasing its profitability and contributing to the reduction of the environmental impact is very important for the sustainability of these small industries [25,26,27].

Therefore, the objectives of this study are as follows: (i) to study the performance of ultrafiltration, followed by diafiltration of retentates, for the production of goat and second sheep whey protein concentrates; (ii) to design a membrane installation for a small/medium size cheese dairy, based on the experimental results obtained at a pilot scale; (iii) to assess its economic profitability through cost-benefit analysis considering that the whey protein concentrates will be used for the development of new products from goat and sheep milk, namely whey cheeses with probiotics, food dairy gels, and whey powders to market.

## 2. Methodology

### 2.1. Sampling and Pretreatment of SCW and GCW

Six samples of goat cheese whey (GCW) and second cheese whey from sheep (SCW) were collected in the same cheese dairy, located close to Beja (Portugal). In this cheese dairy are produced, twice a week, sheep cheese and, in the other three days a fresh goat cheese. When sheep cheese is produced, the cheese resulting is used to make whey cheeses, from which second cheese whey is produced. A volume of about 50 L of each sample was collected and carried out to our laboratory, keeping them refrigerated in ice, during transporting. After arriving, samples were filtrated two times through cotton cloths, similar those used in traditional cheese dairies, to remove suspended solids and casein fines. After that, samples were skimmed in a Elecrem brand centrifuge, at a temperature of about 35 °C, to remove most of the lipids and some minor residues of casein and bacteria. Since these samples have a high concentration of lipids, the reduction of their concentration is crucial to avoid membrane fouling. At finally, samples were subjected to a low pasteurization, at 65 °C, for 30 min. When it was not possible to prepare the samples in the same day, they were immediately preserved at about 3 °C, until the next day.

### 2.2. Filtration Experiments

The filtration experiments included the ultrafiltration (UF) of the pretreated goat cheese whey and second cheese whey from sheep (PGCW and PSCW) to separate/concentrate protein fraction from and obtain a lactose-rich permeate. To purify protein fraction, diafiltration/ultrafiltration (DUF) of the retentates was also carried out. After that, all the permeates resulting from UF and DUF were collected for future processing by nanofiltration. It is proposed in this study to evaluate the possible valorization of the whey protein concentrates obtained for production of food gels, whey cheeses with probiotics and protein powders (Figure 1).

Before each permeation test, the new membranes were subjected to a cleaning and disinfection cycle. The cleaning procedure involved rinsing three times to remove membrane preservatives and, after that, the sequential permeation of the following solutions: sodium hydroxide (Fisher Chemical, Loughborough, Leics, UK); ethylenediamine tetra-acetic acid, sodium salt, Titriplex III (Merck, Darmstadt, Germany); nitric acid (Panreac, Darmstadt, Germany) and citric acid-1-hydrate Chem-Lab, (Zedelgem, Belgium), in concentrations and operating conditions of pressure, temperature, and pH, recommended by the manufacturer. After the washing with a cleaning solution, water rinsing was performed to remove the reagent. The disinfection process involved the permeation of a solution of hydrogen peroxide 30% (Carlo Erba Reagents S.A.S., Val de Reuil Cedex, France). Both cleaning and disinfection process occurred at a temperature around 25 °C. The hydraulic permeability to pure water was determined by measure of permeate fluxes at different transmembrane pressures, a feed circulation velocity of 0.94 m s^−1^ and a temperature of 25 °C. The hydraulic permeability to pure water is the slope of the linear regression obtained from the experimental water fluxes and corresponding transmembrane pressures (1)
(1)$$J_{w}=\left(\frac{L_{p}}{\mu }\right)\Delta P$$
where $J_{w}$ is the water permeate flux (m s^−1^); $\left(L_{p}/\mu \right)$ is the hydraulic permeability to pure water (m s^−1^ Pa^−1^); $L_{p}$ is the intrinsic permeability of the membrane (m), related with its morphological characteristics; μ, the water viscosity (Pa s) and ΔP, the applied transmembrane pressure (Pa).

After the tests, membranes were cleaned with an alkaline solution at 0.01% (Ultrasil 11, Ecolab) followed by disinfection with a solution of hydrogen peroxide, 1000 mg L^−1^, at a temperature of 25 °C. To ensure that membrane’s permeability characteristics were maintained, the hydraulic permeability to pure water was again determined and, if it was at least 95% of its initial value, the same membranes were used in the following tests. If hydraulic permeability of membranes was not recovered, then a complete cycle of cleaning and disinfection, like that used for the new membranes, was realized.

#### 2.2.1. Ultrafiltration and Dia/Ultrafiltration Experiments

Ultrafiltration experiments were carried out with organic membranes with an active layer made of regenerated cellulose acetate and a molecular weight cut-off of 10 kDa, designated by RC70PP. Membranes were furnished by Alfa Laval, Navskov (Denmark). The membranes and optimized operating conditions of transmembrane pressure (0.2 MPa) and feed circulation velocity (0.94 m s^−1^) were selected in previous works [28]. The temperature changed from around 15 °C until 17 °C and all the permeate fluxes values were converted for the same temperature of 25 °C, according with the relationship between fluxes and temperature [29]. The equipment used was a plane and frame module, from Alfa Laval, Navskov (Denmark), suitable for microfiltration, ultrafiltration, nanofiltration, and reverse osmosis processes.

Ultrafiltration experiments, with both type of samples, were done in three steps: pre-concentration until the desired volume concentration factor, VCF (volume of feed/volume of retentate) = 4.00; dilution (diafiltration mode) by adding deionized water and post-concentration. This procedure allows to achieve the concentration/purification of protein fraction in the retentate and a better recover of lactose in the permeate, thus contributing to improving the separation between these components.

Starting from an initial volume of 40 L of each sample, UF experiments were carried out in a batch operation with full recycle of retentate until a volume concentration factor (VCF) of about 4, keeping a membrane area of 0.144 m^2^. After this preconcentration step, diafiltration (DF) of the final retentates was performed, in three stages in a discontinuous mode. In each of them, a volume of deionized water, equal to the observed volume of the retentate in the tank, that is a diavolume, DV (volume of water added/volume of retentate) = 1, was added in the beginning of the dilution process. After homogenization and stabilization of the same operating conditions of transmembrane pressure, feed circulation velocity and temperature, during about 20 min, a new concentration process took place, being the final retentate formed in the feed tank over time, until the same volume of permeate was collected, thus maintaining the volume of the retentate constant. All the processes UF and dilution mode (DF) were realized in discontinuous mode.

#### 2.2.2. Evaluation of the Performance of UF and UF/DF Processes

The performance of the process was assessed through the determination of the following parameters: permeation fluxes, *J_p_*, and selectivity or separation factor, α to separate protein and lactose fractions.

Along the experiments, the volumetric permeate fluxes were determined according to Equation (2)
(2)$$J_{p}=\frac{\Delta V}{A_{m}\times \Delta t}=\frac{q_{p}}{A_{m}}$$
where $J_{p}$ is the volumetric permeate flux; ΔV, the volume of permeate collected during an interval of time Δt (s), and $A_{m}$ is the total membrane area (m^2^).

Samples of raw and pretreated GCW and SCW, retentates and permeates from UF and DF were taken for physicochemical analyses and determination of the separation factor, α, between macrosolutes (protein) and microsolutes (lactose) defined as [30].
(3)$$\alpha =\frac{S_{microsolute}}{S_{macrosolute}}$$
where: $S_{microsolute}$ is the sieving coefficient for the microsolute (lactose) and $S_{macrosolute}$ is the sieving coefficient for the macrosolute (protein). The sieving coefficient for a component is defined as: Si = c_p_/c_r_, being c_p_ the concentration of a solute in the bulk permeate and c_r_, the concentration of the solute in the bulk retentate. The greater the separation factor between microsolute and macrosolute, the more efficient the separation.

### 2.3. Physicochemical Characterization of the Samples

The feed, retentates and permeates from UF and UF/DF were analyzed for: pH (by potentiometry); lactose, according to the method described in [31]; total solids, by gravimetry [32]; ash, by incineration, at 550 °C in a muffle furnace; total nitrogen, by the Kjeldahl reference method and crude protein, obtained from total nitrogen multiplied by the factor 6.38 [33] and adapted for cheese whey, by using about 2.5 g of sample and 10 mL of concentrated hydrochloric acid 0.098N and a catalyst mixture composed of copper and potassium sulphates. The fat content was determined by infrared spectroscopy using the equipment Milkoscan134B, previously calibrated for cheese whey with the standard method of Rose-Gottlied for milk and dairy products.

### 2.4. Design of a Membrane Facility to Recover Protein Fraction of GCW and SCW

The design of the membrane installation will include the determination of the membrane area, after setting a desired feed flow rate and the calculation of the energy consumed. In addition to these variables, the following parameters are also estimated, based on data from the literature and/or on the experimental results obtained: lifetime of the membranes; daily cleaning time; manpower; volume of water required for diafiltration operations; losses of permeate and amount of chemicals for cleaning operations.

#### 2.4.1. Scaling of Membrane Area

In this study, the design of a membrane installation to recover the protein fraction of cheese whey for further valorization, will be carried out using ultrafiltration, followed by diafiltration in three stages. A membrane facility working in a batch mode and with full recycle of retentate for ultrafiltration or diafiltration (when dilution water is used) is shown in Figure 2.

For preconcentration by UF, there is no water added. A volumetric balance at the flow rates in membrane module can be written as
(4)$$q_{f}=q_{r}+q_{p}$$
where $q_{f}$, $q_{r}$, and $q_{p}$ are the feed, retentate and permeate flow rates, respectively.

The volume concentration factor (VCF) is defined as
(5)$$\mathrm{VCF}=\frac{q_{f}}{q_{r}}$$
with the experimental values of permeate fluxes, obtained from Equation (2), the average permeate flux, $J_{av},$ along the concentration process, in a batch mode, can be estimated as [9]
(6)$$J_{av}=0.5\;\left(J_{o}+J_{f}\right)$$
where $J_{0}$ and $J_{f}$ are the initial permeate flux and the permeate flux at the end of the concentration process, respectively.

For a desired VCF and using Equations (4) and (5) it is possible to obtain the permeate flow rate. Then, substituting in Equation (2) and using the value of the average permeate flux (6), membrane area to process a feed flow rate of 450 L h^−1^ can be calculated. The same procedure was adopted for sizing the diafiltration system, in three stages.

#### 2.4.2. Estimation of Energy Consumption

The energy consumed in the UF/DF process has usually three main inputs: thermal energy, that should be considered when heating or cooling is required to keep a desired temperature; energy for feed pumping, E_f_, to feed the system at the adequate transmembrane pressure and energy for feed recirculation, E_Q_, for keeping the required flow rate.

In this work, thermal energy term is not considered since, during experiments, heat was not added and cooling was only controlled through the water flow rate provided to the heat exchanger, to control the temperature. The electrical energy required for feed pumping, E_f_, is proportional to the feed flow rate and transmembrane pressure, and is estimated by Equation (7) [9,30]
(7)$$E_{f}=\frac{P_{f\;}\times q_{f}}{1000\;\eta }\times \Delta t\;\left(\mathrm{kwh}\right)$$
where $P_{f}$ is feed pump pressure (Pa); $q_{f}$ is feed flow rate (m^3^ s^−1^); Δt=1 h and η , pump efficiency (0.7) [9].

The electrical energy necessary for feed recirculation is proportional to pressure drop across the module (experimentally determined) and to feed recirculation flow rate and is calculated by Equation (8) (adapted from [9])
(8)$$E_{Q}=\frac{\dot{\Delta P}\times Q}{1000\;\eta }\times \Delta t\;\left(\mathrm{kWh}\right)$$
where $\dot{\Delta P}$ is the pressure drop across the module (Pa); Q is the feed recirculation flow rate (m^3^ s^−1^); Δt=1 h and η, pump efficiency (0.7).

### 2.5. Proposal of the Methodology for Development of the New Products from LWPC

The methodology proposed in this work for further development of the new products from LWPC is below indicated.

The production of dried LWPC, with 60% or 80% of crude protein, will be carried out by evaporation of the LWPC, followed by drying [9].

The production of whey cheeses with probiotics will be performed through thermal precipitation of whey proteins. This process will include heating the cheese whey under smooth stirring conditions, in a range of temperature 90–100 °C, during about 15–30 min, for whey protein precipitation. After that, the precipitated fraction is separated from the remaining liquid, cooled and mixed with probiotic cultures, according with a process like that described in [34]. Although the production of whey cheeses is usually carried out in small or medium size cheese dairy plants, especially from ovine cheese whey, and this product is much appreciated, its shelf life is short, about four days, in refrigeration conditions. Therefore, the development of this new product, with addition of probiotics can contribute to increase the shelf life of this new product, in addition to its beneficial properties for health, as described in Section 1.

The production of gels can be induced by heat, at a temperature of about 90 °C, for 5 min, or by lowering the pH by adding glucono-δ-lactone (GDL) leading to acid induced gels that can be commercialized for incorporating in various products, such as mayonnaises, yoghurts, desserts, giving them improved functional properties [20,21,24].

### 2.6. Cost–Benefit Analysis (CBA)

In this study, the CBA technique is used to assess the profitability of installing an industrial membrane unit in a small/medium sized cheese dairy in “Baixo Alentejo” region (Portugal), which produces an average of 3500 L of goat’s and second sheep’s cheese whey, every other day, without any commercial utility. The CBA includes the calculation of three indicators [35]:

Net Present Value (NPV), that is, the discount value of the sum of costs and benefits which occur in the lifetime considered, calculated according with the equation
(9)$$\mathrm{NPV}=-{\mathrm{CF}}_{0}+\sum_{i=1}^{n}\frac{{\mathrm{CF}}_{i}}{{\left(1+r\right)}^{i}}$$
where CF_0_ is the project’s initial investment; n, is the total number of years, for which the project was appraised; i, is the time in years; CF_i_ is the future net cash flow of the project; r, is the discount rate.

Internal rate of return (IRR), that is, the annual percentage rate return on investment, determined in accord with the equation
(10)$$0=\mathrm{NPV}=\sum_{i=1}^{n}\frac{{\mathrm{CF}}_{n}}{{\left(1+\mathrm{IRR}\right)}^{i}}{-\;\mathrm{CF}}_{0}$$
where CF_0_ is the project’s initial investment; n, is the total number of years, for which the project was appraised; i, is the time in years; CF_n_ is the net cash inflow during the period n; IRR is the internal rate of return.

Payback period (PBP), that is, the number of years after which total revenue first equals (or exceeds) the total costs, determined according with the following equation
(11)$$\sum_{i=0}^{\mathrm{Payback}}\frac{{\mathrm{CF}}_{i}}{{\left(1+r\right)}^{i}}\;\geq \;0$$
where CF_i_ is the future net cash flow of the project; i, is the time in years and r, is the discount rate.

Data collected from various companies that sell equipment for the dairy industry were used, and national and regional price lists for consumables or labor.

## 3. Results and Discussion

### 3.1. Physicochemical Characterization of Samples

The proximate physicochemical characterization of the following samples: raw SCW and GCW; pretreated SCW and GCW; liquid whey protein concentrates from UF, LWPC (UF), and whey protein concentrates from UF/DF, LWPC (UF/DF), and UF permeates is shown in Table 1 and Table 2, for SCW and GCW, respectively.

The results presented in Table 1 and Table 2 shows that the composition of raw samples is quite different in terms of protein and fat contents, being the concentrations of these components almost similar in the case of GCW, which does not happen in the case of SCW, where the concentration of fat is about a half of that of protein. This result is because GCW is a by-product of goat cheese manufacture, while SCW is a by-product of whey cheese manufacture and so, most of fat was already retained both in sheep cheese and whey cheese. Besides, in goat’s milk, the absence of agglutinin allows for a better dispersion of fat, thus making the skimming process more difficult. This process should be optimized, in our future work, to achieve a better removal of fat, which could lead to an improvement of UF process.

In the relation to the composition of pretreated samples, a huge difference is also observed because the pretreatment carried out was very effective for SCW, because removal of fat was almost complete and no losses of crude protein occurred, unlike what happened with GCW. In fact, in this case, only about 50% of fat was removed, despite defatting have been carried out two times, and around 30% of crude protein was lost during pretreatment. Therefore, until the VCF = 4.0, it was possible to obtain by UF, LWPC of about 32% and 22% of crude protein, in a dry basis, from SCW and GCW, respectively. After a three stage diafiltration, the removal of the lower molecular components, such as lactose and minerals, into permeates was facilitated, thus leading to the purification of protein concentrates, yielding LWPC’s (UF/DF) with about 79% and 64% of crude protein from SCW and GCW, respectively.

### 3.2. Assessment of UF/DF Performance

The average permeate flux of UF experiments varied from around 83 L h^−1^ m^−2^, at the beginning of the experiments, until about 54 L h^−1^ m^−2^, where the desired VCF of 4.0 was reached, which corresponds to a decrease in permeate flux of approximately 35%. This affect the productivity of the process and can be attributed to some fouling on membranes, caused by the accumulation of rejected solids near the membrane surface, especially proteins, residual lipids, mainly phospholipoproteins from the fat globule membrane [36,37] and/or to the formation of insoluble calcium phosphate salts, that partial blocked membrane pores. However, after a cleaning and disinfection cycle, more than 98% of water hydraulic permeability was recovered and the same membranes were used in the following UF/DF experiment. During diafiltration of UF retentates in three stages, permeate fluxes were kept constant, indicating that no fouling occurred and therefore low molar weight components, mainly lactose and salts, could easily permeate UF membranes.

The results of the separation factor between crude protein and lactose fractions along the UF/DF process are presented in Figure 3.

As can be observed in Figure 3, the separation between protein and lactose for UF/DF of SCW is much higher than the one obtained with GCW, being in the range 8.2–25.5, while for GCW it varies between 3.4 and 8.0. This great difference can be probably due to the fact that in GCW, crude protein and lipids concentration is at a similar level, probably leading to fouling by lipids, and, consequently to the increase of apparent rejections of lactose (from about 5% until 23%), impairing the selectivity of the separation process. Therefore, it is advisable to improve the pretreatment used for the removal of lipids from GCW, for a better performance.

The discontinous diafiltration process of retentates in three stages, proved to be a good strategy to significantly increase the separation between whey proteins and lactose, which allowed to increase protein fraction purification, which benefits its functional properties, broadening its range of applications [15]. However, some of the constraints of diafiltration processes are time, water and energy consumption and required membrane area. Based on the experimental results obtained, a proposal for the instalation of a UF/DF plant in a small or medium size cheese dairy plant is presented in the following sections.

### 3.3. Design of UF/DF Membrane Unit

The way in which the installation works is determines the performance of membrane operations. Based on the experimental results collected in batch mode, in a plate and frame module, for UF and UF/DF of both CGW and SCW, a proposal for the installation of a membrane facility in a small/medium size cheese dairy plant is shown in Figure 4.

For this process, it is necessary a storage tank with the capacity to contain the volume of GCW/SCW produced daily and a feed tank, with enough capacity to process the estimated volume of pretreated GCW or SCW (450 L h^−1^). This feed flow rate is an average value, estimated based on the results obtained from surveys of cheese producers in 45 cheese dairy plants, in the Baixo Alentejo region, Portugal (results not shown). The feed tank is associated with the membrane module, so that the retentate can pass the required number of times through the membranes, until the desired VCF is reached. Then, the obtained LWPC (UF) will be transferred to tank 8, where DF process will be carried out, in three stages. Therefore, each batch will proceed in four stages, one for preconcentration by UF, and three stages for UF/DF using the same module. As soon as the LWPC (UF) is transferred to tank 8, a new volume of pretreated cheese whey will be introduced by the same process. As can be seen in Figure 4, for the operation of this configuration three pumps are required, one responsible for the transport and pressurization of the feed and two for retentate flow recirculation, through the membrane module. This configuration adjusts to different needs for the volume of cheese whey to be processed, being able to be adapted both for small and medium/size cheese dairy plants. Compared with traditional continous plants using co-current processing, batch processing can offer several advantages, such as: less membrane required and lower volume of diafiltration water to achieve the same purification degree; the filtration module is more compact and fits in a smaller space, it is easier to clean and sterilize and permeate quality can be controlled at the end of the process and so it can be improved by total or partial second-pass treatment as necessary [38]. One of the constraints for the use of systems operating in batch mode, compared with continous process, is the required tankage [9], although at small scales this factor may not be relevant.

#### Technical Evaluation of UF/DF Membrane Unit

The results obtained to estimate the required membrane area to process 450 L h^−1^ of GCW or SCW are presented in Table 3.

The use of a preconcentration by UF until a VCF of 4.0, in one simple stage, followed by three stages of DF, each one until a VCF = 1.5, allow to obtain goat’s and sheep’s whey protein concentrates of about 60% and 80%. However it would be important in our future work to carry out UF and UF/DF tests up to higher VCF to be able to define the optimal VCF for the recovery of the protein fraction and minimize volume of water added and saving time. In addition, performing the same tests but with spiral-wound modules, the most used at an industrial level, could help to make an assessment closer to reality. The evaluation of the volume of UF permeate generated, the volume of diafiltration water used, energy consumption, and other technical parameters are shown in Table 4.

The data presented in Table 4 allow to conclude that for the processing of 450 L h^−1^ of GCW or SCW, about 115 L of LWPC (UF/DF) with a protein concentration between 60% and 80% are obtained. These, will be valorized for production of whey cheeses with probiotics, food gels, or dry powders. The total permeate volume, that includes the permeate from UF and those of DF operations, is aproximately 500 L. One of the possibility to reduce the volume of final permeate is to use the permeates from DF3 as diafiltration water, in the first stage of diafiltration, as described in [28].

The energy for pumping and recirculation of retentates were estimated through Equations (8) and (9). For the calculation of the pumping energy (E_f_), the pressure used was 2.0 × 10^5^ Pa(average pressure used in the UF/DF tests); the feed flow rate 450 L h^−1^ and the pump efficiency, 0.7 [9,30]. It was considered the pumping energy for UF and diafiltration experiments. Ultrafiltration experiments will be carried out in a module of an area of 4.5 m^2^ and the three diafiltration stages in another module of an area of 1 m^2^. To determine the energy consumed by the recirculation pump (E_Q_), the pressure loss obtained experimentally was taken and an experimental recirculation retentate flow of 10 L min^−1^ per module was considered, for a efficiency of the recirculation pump of 0.7. The results of energy shown in Table 4 are referred to a working day in the cheese factory, which is usually 8 h. The results obtained are in accordance to what usually happens in ultrafiltration, where energy for pumping is lower than that of recirculation, because of the lower transmembrane pressures used. The lifetime of membranes was arbitrarily assigned, considering the operation of the membrane installation for a year, working 8 h a day. The amount of cleaning and disinfection chemicals used was estimated considering that one cycle of washing and disinfection will be realized, by day.

### 3.4. Cost–Benefit Analysis

This section intends to evaluate the economic feasibility of installing an industrial membrane unit for the reuse of second sheep’s and goat’s cheese whey, resulting from the manufacture of goat cheese and whey cheeses from sheep. The analysis has as reference a small/medium-sized cheese dairy, which produces an average of 3500 L of GCW and SCW daily, without any commercial utility.

This installation has a strong innovative aspect, since it involves the development of new products that can diversify the cheesemaking activity, eventually increasing its profitability and contributing to the reduction of the environmental impact of the activity in question.

#### 3.4.1. Sales Revenue

As part of the installation of membranes, it is intended to develop three new products (food gels; whey cheeses with probiotics and protein powder concentrates), obtained from LWPC of GCW and SCW, using membrane technology. Food gels will be sold in 0.5 kg packages. The whey cheese with probiotics and the protein powder concentrates (protein concentrations of 60% and 80%) will be sold by weight to end customers (individuals and food industries). For a feed flow rate of about 3500 L of GCW and SCW daily, it is possible to produce 10,000 kg of food gels, 3000 kg of whey cheeses with probiotics and 3000 kg of protein powders per year. An estimate of the unit sales prices and sales of these products obtained through a survey in the market is shown in Table 5. It was considered that the average sale price of the products will rise by 2% per year, due to inflation and the increase in price for brand awareness.

#### 3.4.2. Investment and Financing

Data collected from several firms that sell equipment for industry show that remodeling of facilities and equipment necessary to incorporate an industrial membrane unit in the cheese dairy plant points to an initial investment of 172,000 € (year 2021). The equipment identified as necessary, and the respective prices are (1) complete membrane equipment, including tank, feed pump, one spiral-wound membrane module, heat exchanger, permeate collectors: € 30,000; (2) recirculation pumps: € 10,000; (3) remodeling of installations: € 20,000; (4) 3 stainless steel tanks: € 5000; (5) remodeling of the CIP system: € 4000; (6) spray-dryer unit: € 100,000; (7) miscellaneous equipment: € 3000.

The planned investment can be partially financed by applying for existing Community funds. The support programs in force in the Alentejo Region make it possible to estimate that 50% of the value of investment will be obtained with investment grant. In the example presented, the remaining 50% of the investment is obtained through bank financing. According to consulted bank sources, it will be possible to obtain a five-year loan from the bank with an interest rate of 4.5%.

#### 3.4.3. Operational Expenses

The current annual expenses estimated with the operation of the membrane unit, are shown in Table 6 and refer to 2021. Data were collected from several companies that sell products for the dairy industry with respect to various inputs. Portuguese Energy Services Regulatory Authority and water supply companies were consulted to obtain the prices of water and energy for the industry, respectively € 0.083/kWh and € 0.16/m^3^. This table shows the expenses in two cost centers (Industrial Membrane Unit, Spray Drying Unit, and Pre-treatment). Data on national and regional prices with respect to various consumables or labor costs were collected from listings by the Portuguese statistical authority (INE). The different costs are updated over the years of the analysis, using the annual inflation rate of 2%. This is calculated based on the evolution of the values recorded in the last five years in Portugal. The implantation of the membrane industrial unit in the cheese dairy plant implies the reinforcement of the company’s employees and the following additional labor costs: (1) food engineer (1/4 time) € 10,000; (2) commercial assistant (1/4 time) 6000 €; (3) administrative assistant (1/4 time) 4000 €; (4) two machine operator (1/2 time) € 14,000; (5) other labor costs € 1500.

It was estimated that the commercial costs of the dairy will also increase by € 7000/year due to the growth in advertising expenses (€ 5000) and other commercial expenses (€ 2000).

#### 3.4.4. Other Information for CBA Calculation

The value of the non-repayable subsidy was spread over five years, in accordance with accounting standards. To estimate the depreciation amounts, the regulatory framework in force in Portugal was used (R. D. 25/2009). Considering the specific rates for “food, beverages and tobacco”, an average rate of 20% was used to calculate depreciation and amortization.

Regarding the repayment of the loan and the respective interest, a bank financing of € 90,000 over five years was considered, with an interest rate of 4.5% (established after consultation with three credit institutions). A credit simulator was used to obtain the amount of the monthly installment to be paid during the five years. Thus, the monthly amortization of the loan amounts to € 134,037 and the interest to be paid monthly will be € 33,750/month, totaling € 20,134 per year (principal and interest). A tax rate of 20% was used, considering that the tax rate on corporate income in Portugal is 17% or 21%, depending on the tax base. Depreciation compensation refers to the amount recorded as an expense related to depreciation that does not give rise to payments. It was estimated that the residual value of the project assets would be equal to 10% of the investment value.

To calculate the net present value (NPV), a required rate of return of 15% was considered. This rate of return (or discount rate) is no more than a risk-free interest rate plus a risk premium established for the type of project concerned.

The positive environmental effects resulting from installing an industrial membrane unit in a small-sized cheese dairy are considered in terms of cost reduction for society. These benefits have not been quantified (and therefore, it will not be considered in the CBA) due to the lack of reliable data or the impossibility of objectively estimating its effect.

#### 3.4.5. Financial Cash Flow Statement

Table 7 shows the cash flow statement which includes the revenues generated from the adaptation of an industrial membrane unit in a small and medium-sized cheese dairy, the annual operating costs, and the total investment cost. The cash flow statement was built to see the return the owner would have on investing in the project.

#### 3.4.6. CBA Indicators Results

The result of the financial analysis of installing an industrial membrane unit in a small-sized cheese dairy shows that at a financial discount rate of 15%, the project will generate a positive net financial present value of € 1,602,681. The analysis also allows to conclude that the project will generate an IRR of 21% higher than the discount rate, which means that the project will have a higher rate of return than the opportunity cost of investing in an alternative project. The repayment period indicates that in approximately three years and three months the total revenue first equals the total costs.

#### 3.4.7. Sensitivity Analysis

Sensitivity analysis measures the sensitivity of the result of the project to changes in one parameter’s value at a time. We redid the calculations of NPV, IRR, and PRP in two pessimistic scenarios:

C1: where sales are maintained over the five years (this means a real decrease equal to the value of inflation) and C2: with a generalized increase in labor costs of 10%. In scenario 1, the project will generate a positive NPV of € 671.21 and has an IRR slightly higher than the discount rate (15%). In scenario 2, the VPN is € 3,099.88 and the IRR is 16%.

The repayment period in scenario 1 is approximately five years while in scenario 2 is four years and 11.5 months. These results reveal that the impact of the two scenarios on the project outcome can be significant, although the indicators are still acceptable. Nevertheless, the need for attention by the management is evident so that they can be mitigated.

## 4. Conclusions

The sequential process UF, followed by UF/DF of retentates led to the valorization of both GCW and SCW, through the production of LWPC, which can be used to develop new and interesting products from the milk of these small ruminants. In this study the valorization of UF permeates is not included, that also contains very nutritional components—such as lactose, minerals, vitamins, oligosaccharides—that may be used directly for production of high-value ingredients, through fermentation with selected lactic acid bacteria or by addition of acids. Another solution for recovering these permeates is also to use membrane processes, namely nanofiltration to recover/concentrate lactose and oligosaccharides to commercialize, producing final permeates which can be reused in dairy itself, avoiding at the same time, its environmental impact. Our future work will be focused on the valorization of these permeates.

Cost–benefit analysis and Sensitivity Analysis allowed us to conclude that it will be profitable for a small/medium size cheese dairy plant, with an average daily production of about 3500 L of GCW and SCW, to install a membrane unit to obtain LWPC, from which it is possible to obtain new products (food gels, whey cheeses with probiotics and whey protein powders). However, it would have been important to conduct a market study to get to know the target customer better, as well as industries interested in purchasing these new products, namely food gels.

This analysis, to be applied in cheese dairy plants with lower daily production of those by-products, would have to be re-analyzed, considering, especially the initial investment. A solution that could be applied, in the case of small cheese dairies, would be, for example, the installation of a common membrane unit, where small cheese producers could deliver their by-products for recovery. This collective unit would have great potential to obtain a high investment grant.

## Figures and Tables

**Figure 1 foods-10-01740-f001:**
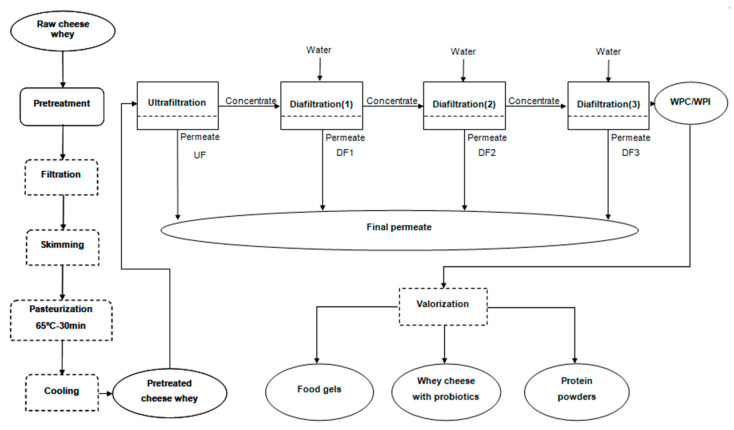
Experimental plane.

**Figure 2 foods-10-01740-f002:**
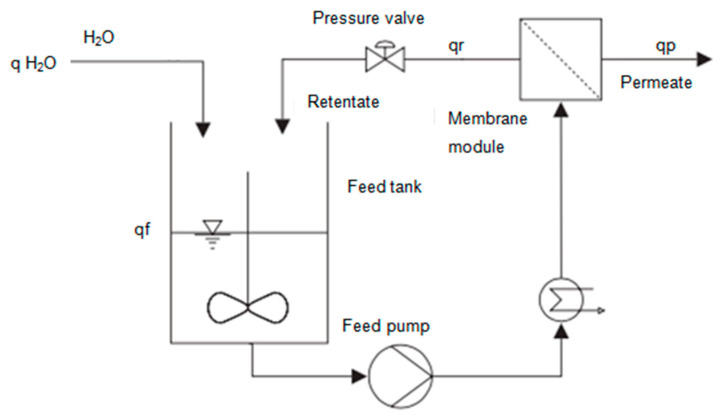
Scheme of a discontinuous operation with full recycle of retentate (adapted from [15]).

**Figure 3 foods-10-01740-f003:**
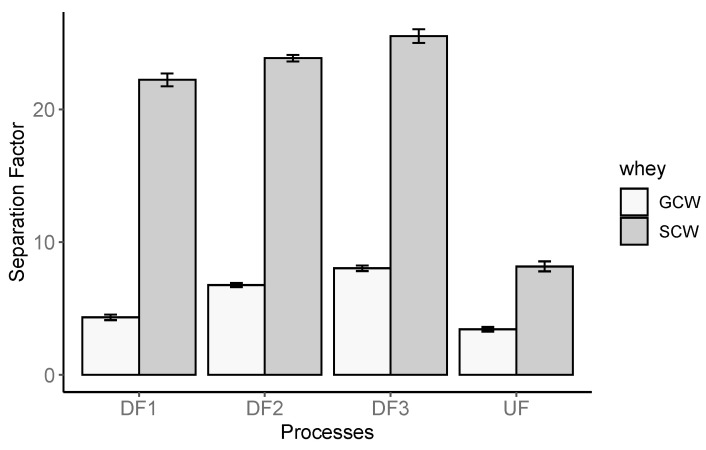
Variation of separation factor along UF/DF process of goat cheese whey (GCW) and second cheese whey (SCW) from sheep, obtained at $\Delta P=2.0\times 10^{5}$ Pa, v = 0.92 m s^−1^ and T = 25 °C.

**Figure 4 foods-10-01740-f004:**
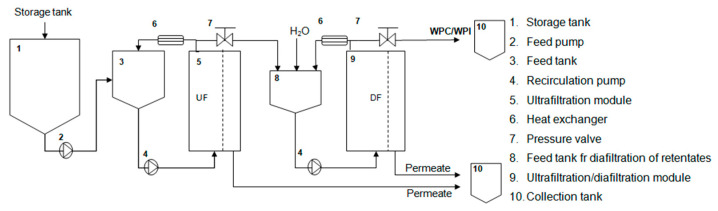
Scheme for the installation of membrane processes to recover GCW and SCW for a small/medium size cheese dairy.

**Table 1 foods-10-01740-t001:** Physicochemical characterization of samples of Second cheese whey from sheep (SCW) (dry basis).

Parameter (% *w*/*w*)	Raw SCW	Pretreated SCW	LWPC (UF)	LWPC (UF/DF)	Ultrafiltration (UF) Permeate
Lipids	6.73 ± 0.34	<0.1 ^(1)^	0.31 ± 0.05	7.73 ± 0.15	<0.1
Crude protein	10.73 ± 1.83	10.93 ± 1.28	31.86 ± 1.75	79.47 ± 1.41	5.35 ± 0.26
Lactose	58.06 ± 7.08	59.49 ± 5.84	45.59 ± 2.39	8.04 ± 0.31	62.41 ± 0.98
Ash	28.54 ± 1.19	30.69 ± 2.39	23.52 ± 1.59	6.61 ± 0.72	30.18 ± 0.19

^(1)^ Detection limit. LWPC: liquid whey protein concentrates

**Table 2 foods-10-01740-t002:** Physicochemical characterization of samples from Goat cheese whey (GCW) (dry basis).

Parameter (% *w*/*w*)	Raw GCW	Pretreated GCW	LWPC (UF)	LWPC (UF/DF)	UF Permeate
Lipids	10.76 ± 1.83	5.73 ± 0.25	7.00 ± 0.34	12.18 ± 2.00	9.68 ± 0.42
Crude protein	10.39 ± 1.24	7.35 ± 0.34	21.91 ± 0.17	63.62 ± 6.75	6.24 ± 0.24
Lactose	55.38 ± 6.23	63.01 ± 6.80	51.92 ± 6.70	17.66 ± 0.17	64.95 ± 0.80
Ash	23.47 ± 0.20	23.91 ± 0.20	18.94 ± 0.15	6.54 ± 0.24	28.28 ± 0.20

**Table 3 foods-10-01740-t003:** Process parameters for *q_f_* = 450 L h^−1^ of GCW or SCW; ΔP=2.0  bar; Qrec = 10 L min^−1^.

Process	Stage	VCF/Stage	J_p(average)_ (L h^−1^ m^−2^)	A_m_ (m^2^)
Preconcentration by UF	1	4.0	76.6	4.5
DF1 (DV = 1.0)	1	1.5	58.9	1.0
DF2 (DV = 1.0)	2	1.5	58.4	-
DF3 (DV = 1.0)	3	1.5	55.5	-

DV = (V_water added_/V_retentate_); VCF = volume concentration factor; J_p_ = permeate flux; A_m_ = membrane area.

**Table 4 foods-10-01740-t004:** Technical parameters of UF/DF membrane plant for *q_f_* = 450 L h^−1^ of GCW or SCW.

Parameter	Value
Permeate flow rate (L h^−1^)	330
Retentate flow rate (L h^−1^)	115
Total membrane area for UF and UF/DF (m^2^)	5.5
V_diafiltration water_ (L)	170
E_f_ (kWh) (for a working day of 8 h)	0.48
E_Q_ (kWh) (for a working day of 8 h)	1.28
Lifetime of membranes (h)	2000
Cleaning time (h day^−1^)	1–2
Cleaning chemicals (kg)	0.1
Manpower (h day^−1^)	1
Cheese whey losses in membrane plant (%)	1

**Table 5 foods-10-01740-t005:** Annual sales forecast.

Products	Unit Price	Sales (Amount)	Sales (Value)
Food gels	4 €/(package 0.5 kg)	20,000 package	80,000 €
Whey cheese with probiotics	8.50 €/kg	3000 kg	25,500 €
Protein powder concentrates (60% protein)	4 €/kg	1500 kg	6000 €
Protein powder concentrates (80% protein)	4 €/kg	1500 kg	6000 €
Total	-	-	117,500 €

**Table 6 foods-10-01740-t006:** Operating costs (Euros).

Operating Expenditures	Industrial Membrane Unit	Spray Drying and Pre-Treatment	Total (Euros)
Membranes (m^2^)	1000		1000
Energy	3000	4000	7000
Specialized services (maintenance)	5000	1200	6500
Water	1200		1200
Insurance	300	200	500
Cleanliness, hygiene (imputation)	1000	1000	2000
Various packaging	4000		4000
Other external supplies and services	1000	1000	3000
Total	16,500	7400	23,900

**Table 7 foods-10-01740-t007:** Financial cash flow statement (Euros).

	Year 1	Year 2	Year 3	Year 4	Year 5
1. Sales revenue	117,500	119,850	122,247	124,692	127,185
2. Investment grant	17,200	17,200	17,200	17,200	17,200
3. Operating expenditures	23,900	24,378	24,866	25,363	25,870
4. Labor costs	35,000	35,700	36,414	37,142	37,885
5. Comercial costs	7000	7140	7283	7428	7577
6. Income before depreciation, financing expenses and taxes	14,266	15,298	16,350	17,425	18,519
7. Depreciation	34,400	34,400	34,400	34,400	34,400
8. Operating income	34,400	35,432	36,484	37,559	38,653
9. Installment (Capital + Interest)	20,134	20,134	20,134	20,134	20,134
10. Income before taxes	14,266	15,298	16,350	17,425	18,519
11. Taxes	2853	3060	3270	3485	3704
12. Net income	11,413	12,238	13,080	13,940	14,815
13. Depreciation compensation	34,400	34,400	34,400	34,400	34,400
14. Cash flow exploration	45,813	46,638	47,480	48,340	49,215
15. Investment	−172,000				
16. Residual value					17,200
17. Working capital	−4000				

## Data Availability

www.lacties.com.

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
