# Peer review of "Integration of Membrane Processes for By-Product Valorization to Improve the Eco-Efficiency of Small/Medium Size Cheese Dairy Plants"

_foods, 2021, doi:10.3390/foods10081740_

Round 1

Reviewer 1 Report

This manuscript aimed at generating data from UF/DF processes used to generate new products from goat cheese whey and second sheep cheese whey. This data was used to perform a cost-benefit analysis. Definitely, there is a great commercial potential for by-products obtained from small ruminants dairy products. Although this manuscript is interesting, I think that a major revision is needed.

Major comments

  1. My first comment is related to the energy calculation.

1.1 There is a mistake in equation 8. Power, in watt (W) = P (Pa) x Q (m3/s) / pump efficiency.  If the authors multiply this value with the duration of the experiments, in hours, we have Wh, divided by 1000 to obtain kWh. Since the authors used Bar instead of Pa, we multiply by 100 000 for the unit conversion. Why did the authors used 2,78 x 10-2 factor ?? They do not have to divide by 3600. We only divide by 3600 when we want to convert J to W.

1.2. Equation 9. The authors do not need the membrane surface area. They only have to determine the pressure drop per module, and multiply the delta P X Q per module by the number of modules. What is the 600 factor here?

1.3 Table 4. There are mistakes in energy calculation. If the pump power is 11 kW, then it is not possible that the authors only consumed a total of 0,15 kWh during all the UF/DF process.

  1. I would have a lot of questions regarding the cost-benefit analysis. Presenting the equations used in the M&M section would help the reader to understand how it was done. Operating costs were estimated for the UF/DF process, but no details are provided regarding the spray-drying (and other processing steps than filtration).

2.2 According to me, many expenditures were underestimated, notably energy or labour costs. In the scenario presented, only one man/woman will be able to perform the process from fresh whey to packaged WPC powder ? And the three products (cheese with probiotics, and WPCs) ?)

2.3 How whey cheese with probiotics is made, or how does the fermentation is performed? Is it a product already available in Portugal? I would have the same question for the “food gel” product. Is that product actually sold in Portugal? The increase of 2% per year, due to inflation, seems optimist. In fact, all the assumptions suggested in the manuscript are “possible”, but at least, a sensibility analysis, or a risk analysis would be important for the most critical aspects. Actually, the cost-benefit analysis was only done for a very optimist scenario.

2.4. Many details are not presented. What are the costs for electricity, natural gas or water? Did the authors considered plant expansion or land acquisition costs to enlarge the cheesemaking plant considering the introduction of the spray-drier?

  1. Some details should revised considering filtration.

3.1. Please revise the equation #1. I strongly suggest reading Ultrafiltration and Microfiltration handbook. The unit of the permeate flux is not correct.

3.2 Equation 7. Mean permeation flux is rather volume of permeate collected / duration of the experiment.

Minor comments

Line 68 : Beja (Portugal)

2.2 Filtration experiments ?

Lines 105-109 : Reference to chemicals used must be provided. For example : alkaline solution (Commercial name, Company, city, Country). What was the temperature of the alkaline cleaning ? No acid cleaning was performed ?

Lines 112-114. Membrane manufacturer ?

Line 120. Define VFC. Number of diavolumes ? Correct the word pos-concentration.

Line 124. It is mentioned at the line 118 that the filtration system allows a maximum membrane area of 0.72m2. How did the authors used 0,144m2 at the line 124 ?

Lines 119-129. I understand that a continuous and a discontinuous was performed in two different experiments, but this is not clearly presented. Please reformulate to improve clarity. For example : Two different ultrafiltration/diafiltration were conducted. Firstly, ______continuous diafiltrafion____. In the second experiment, _____discontinuous diafiltration_______.

Line 132. Permeation flux is not productivity.

Line 133, the symbol ∝ is not defined

Lines 147-150. Si is defined under the equation, but is not in the equation 3.

Section 2.3 M&M section is never written to the future (remove “will”)

Line 263. Crude protein, not protein (revise the other occurrences).

Fig. 3. The title of the figure is incomplete. Please increase the font size. Specify that this is the experiment conducted with discontinuous DF ? Where are the results for the continuous DF ?

Table 3. Jp were identical for GCW and SCW ?

Table 4. Correct the units. Day-1, not dia-1

Author Response

Response to reviewer 1.

Reviewer 2 Report

The specific comments were provided in the text of the manuscript. 

The introduction section should focus on factors influencinh UF/DF process than funcitonal properties of whey proteins, which were not the subject of this work. 

You have worked with plate and frame module, but for financial analysis you consider spiral would. This could affect to a great extent the performance of the membranes. Not to mention about the material the membrane is made of. In case of spiral wound what membrane material would be used?

Author Response

Response to reviewer 2.

Reviewer 3 Report

This article presents an interesting study on Integration of membrane processes for by-products valorization to improve the eco-efficiency of small/medium size cheese dairy plants

Comments:

  • Please correct the experiment diagram, it is not very clear
  • Fig 2. Please correct form , not clear enough
  • What can be the microbiological quality of the products after such long processes
  • How to explain the long membrane separation processes and washing processes, if these are not large scale processes, what will be the economic effect, what will be the cost of production of these valuable ingredients ?
  • Describe the computer program used to create the experimental diagrams?
  • Precise process parameters are missing, mainly temperature
  • On what basis was the possibility of producing probiotic products estimated, please describe this in more detail
  • What is the survival of probiotics during product storage?
  • Well carried out cost analyses

Author Response

Response to reviewer 3.

Round 2

Reviewer 1 Report

I would like to thank the authors for considering my comments and suggestions. They did a very serious revision work. I do not have any supplemental comments. Wishing you succes for the next steps !

Reviewer 2 Report

The recommendations were included. The manuscript was greatly improved.